# Are neural quantum states good at solving non-stoquastic spin Hamiltonians?

Chae-Yeun Park[1*] and Michael J. Kastoryano[1,2,3]

**1** Institute for Theoretical Physics, University of Cologne, 50937 Köln, Germany
**2** Amazon Quantum Solutions Lab, Seattle, Washington 98170, USA
**3** AWS Center for Quantum Computing, Pasadena, California 91125, USA
* chae.yeun.park@gmail.com

February 8, 2021

## Abstract

**Variational Monte Carlo with neural network quantum states has proven to be a promising avenue for evaluating the ground state energy of spin Hamiltonians. Based on anecdotal evidence, it has been claimed repeatedly in the literature that neural network quantum state simulations are insensitive to the sign problem. We present a detailed and systematic study of restricted Boltzmann machine (RBM) based variational Monte Carlo for quantum spin chains, resolving exactly how relevant stoquasticity is in this setting. We show that in most cases, when the Hamiltonian is phase connected with a stoquastic point, the complex RBM state can faithfully represent the ground state, and local quantities can be evaluated efficiently by sampling. On the other hand, we identify a number of new phases that are challenging for the RBM Ansatz, including non-topological robust non-stoquastic phases as well as stoquastic phases where sampling is nevertheless inefficient. Our results suggest that great care needs to be taken with neural network quantum state based variational Monte Carlo when the system under study is highly frustrated.**

# 1 Introduction

Over the past decade, Machine Learning (ML) has allowed for huge improvement not only in traditional fields such as image detection [1] and natural language processing [2] but also in other disciplines e.g. defeating human level in playing games [3,4] and predicting protein structures [5]. ML has been also actively applied for solving quantum physics problems such as detecting phase transitions [6,7] and decoding quantum error correcting codes [8–10]. However, arguably the most active contribution of ML to physics has been in the field of classical variational algorithms for solving quantum many body systems so called variational quantum Monte Carlo (vQMC).

A seminal study by Carleo and Troyer showed that the complex-valued restricted Boltzmann machine (RBM) [11] solves the ground state of the transverse field Ising and the Heisenberg models to machine accuracy. Subsequent studies also demonstrated that other neural network based *Ansätze* such as convolutional neural networks (CNNs) [12,13], and models with the autoregressive property [14,15] also show highly accurate results when used with proper optimization algorithms. Despite these successes, the methods still suffer several difficulties in solving highly frustrated systems [12,16,17].

Why then are some Hamiltonians so difficult to solve? In path integral Monte Carlo, it is well known that stoquastic Hamiltonians – those with real-valued and non-positive off-diagonal elements – are tractable [18]. One of the crucial properties of stoquastic Hamiltonians is that the ground state is positive up to a global phase. As the RBM Ansatz also seems to solve stoquastic Hamiltonians well [11], a complex sign structure of the ground state can be a reason why it is difficult to solve highly frustrated Hamiltonian that has strong non-stoquasticity. Thus some recent studies have investigated a relation between signs of the ground state in the computational basis and a difficulty for solving the Hamiltonian using neural quantum states [13,17]. However, the results so far are inconclusive. Even though it has been observed that neural quantum states do not learn sign structures well, it is not clear whether it is intrinsically impossible to represent such a state using a reasonable number of weights, or whether algorithms such as stochastic reconfiguration just fail to find an accurate optimum.

In this work, we investigate the failure cases more systematically and show that non-stoquasticity of the Hamiltonian can cause both problems, but in a distinguishable manner. We identify three typical failure mechanisms: (i) **Sampling**: The sampling method, such as local update Markov chain Monte Carlo (MCMC), fails to produce good samples from the state, or the observables in the optimization algorithm cannot be accurately constructed from a polynomial number of samples. (ii) **Convergence**: The energy gradient and other observables involved in the optimization (e.g. the Fisher information matrix) can be accurately and efficiently obtained for each epoch, yet the optimization gets stuck in a local minimum or a saddle point. (iii) The **expressivity** of the Ansatz is insufficient: the Ansatz is far from the correct ground state even for the optimal parameter set, i.e. $\min_\theta || |\psi_\theta\rangle - |\Phi_{GS}\rangle ||$ is large where $|\psi_\theta\rangle$ is a quantum state that Ansatz describe for a given parameter set $\theta$.

Specific pairs of Hamiltonian and Ansatze can sometimes rule out one or several of the failure mechanmisms. For instance, when the Hamiltonian has a known exact neural-network representation of the ground state (e.g. cluster state, toric code [19] and other stabilizer states [20,21]), we can discard *expressivity* [case (iii)] as a failure mechanism. On the other hand, models with the autoregressive property [14,15] are free from the MCMC errors as they always produce unbiased samples [1]. The Hamiltonians we consider in this work will typically not have known ground state representations and we mainly use the RBM Ansatz, as it is the best studied neural network Ansatz class and the most reliable performance [22]. Hence all three failure mechanisms can occur.

In this work, we set out to understand what role stoquasticity plays in the success and failure of variational Monte-Carlo with the RBM ansatz. By way of example, we show that non-stoquasticity can cause problems with *sampling* [case (i)], while phase transitions within a non-stoquastic parameter region may yield *expressivity* problems [case (iii)]. In particular, we observe the following features of RBM variational Monte-Carlo for a class of one-dimentional XYZ-type Hamiltonians which exhibit a wide variety of stoquastic and non-stoquastic phases:

1. Complex RBMs with sufficient hidden layers ($\alpha \geq 1$) faithfully represent the ground states of spin Hamiltonians that are phase connected to a stoquastic representation, or that can be transformed into such a Hamiltonian with local Pauli transformation.

2. There exists "deep non-stoquastic phases" that cannot be transformed into a stoquastic form using local (on-site) unitaries and are not phase connected to stoquastic Hamiltonians, and which cannot be efficiently represented by complex RBMs.

3. Sampling is stable along the learning path when the Hamiltonian is stoquastic or phase connected to a stoquastic Hamiltonian. However, converged energies from the vQMC may fluctuate even in this case as the number of samples to correctly estimate observables may scale poorly.

Given that "deep non-stoquastic phases" can be gapped, the second observation also implies that the dimension or gap of the system is not related to the reliability of the method in any straightforward manner. Rather, as for quantum Monte Carlo [18], the stoquasticity of the Hamiltonian is the more essential feature. The third observation is about the sampling problem. This type of difficulty already has been observed in quantum chemical Hamiltonians [23] but, in this paper, we explicitly show that one may encounter such a problem even in solving a one dimensional stoquastic system which one would

---

[1]Still, one may see a sampling error if a target probability distribution cannot be well approximated with a finite number of samples.

expect to be rather simple. Finally, we note that phase connectivity is considered under a specific parameterized Hamiltonian with a certain symmetry.

The remainder of the paper is organized as follows. We introduce the complex RBM wavefunctions and our optimization methods in Sec. 2. We next establish our main observations in Sec. 3 by studying how non-stoquasticity affects the RBM using the one-dimensional XXZ and the $J_1$-$J_2$ models the properties of which are well known. We then further confirm our observations using a specially devised Hamiltonian in Sec. 4 and conclude with final remarks in Sec. 5.

## 2 Restricted Boltzmann machine wavefunctions

Inspired by the recent successes in machine learning, Carleo and Troyer introduced the restricted Boltzmann machine (RBM) quantum state Ansatz class [11], and showed that it can accurately solve the ground states of the transverse field Ising and Heisenberg-XXX models in the variational quantum Monte Carlo (vQMC) framework [11]. For complex parameters $a_i, b_j$ and $W_{ij}$ where $i \in [1, \cdots, N]$ and $j \in [1, \cdots, M]$, an (unnormalized) RBM state is given by

$$\widetilde{\psi}_\theta(x) = \sum_y e^{\sum_{i,j} w_{ij} x_i y_j + a_i x_i + b_j y_j} \tag{1}$$

$$= e^{\sum_i a_i x_i} \prod_j 2 \cosh(\chi_j) \tag{2}$$

where $\theta = (a, b, w)$ is the collection of all parameters, $x = (x_1, x_2, \cdots, x_N)$ is a basis vector in the computational basis (typically the Pauli $Z$ basis), $y = (y_1, y_2, \cdots, y_M)$ labels the hidden units, and the 'activations' are given by $\chi_j = \sum_i w_{ij} x_i + b_j$. We also introduce the parameter $\alpha = M/N$ that controls the density of hidden units and parameterizes the expressivity of the model. In addition, we will write $\psi_\theta(x) = \widetilde{\psi}_\theta(x) / \sum_x |\widetilde{\psi}(x)|^2$ to denote the normalized wavefunction.

For a given Hamiltonian, the parameters of the Ansatz can be optimized using a variety of different methods, including the standard second-order vQMC algorithm known as Stochastic Reconfiguration (SR) or a modern variant of the first order methods [15, 24, 25] such as ADAM [26]. Throughout the paper, we use the SR as it is believed to be more stable and accurate for solving general Hamiltonians [27]. At each iteration step $n$, the SR method estimates the covariance matrix $S$, with entries $S_{i,j} = \langle O_i^* O_j \rangle - \langle O_i^* \rangle \langle O_j \rangle$, and the energy gradient $f = \langle E_{\text{loc}}^* O_i \rangle - \langle E_{\text{loc}}^* \rangle \langle O_i \rangle$ where $O_i(x) = \partial_{\theta_i} \log[\widetilde{\psi}_\theta(x)]$ and $\langle \cdot \rangle = \sum_{x \sim |\psi_\theta(x)|^2}(\cdot)$ is the average over samples (see Refs. [11, 27] for more detail). The parameter set is updated as $\theta_{n+1} = \theta_n - \eta_n S^{-1} f$. In practice, a shifted covariance matrix $S' = S + \lambda_n \mathbb{I}$ with a small real parameter $\lambda_n$ is used for numerical stability. In the SR optimization scheme with complex RBM, expectation values are obtained by sampling from the distribution $|\psi_\theta(x)|^2$, typically by conventional Markov chain Monte Carlo (MCMC). In some cases, we use the running averages of $S$ and $f$ when it increases the stability (i.e. we use $f_{n+1} = (1 - \beta_1) f_n + \beta_1 f$ , $S_{n+1} = (1 - \beta_2) S_n + \beta_2 S$ for suitable choices of $\beta_1, \beta_2$ and update $\theta$ using $\theta_{n+1} = \theta_n - \eta_n S_{n+1}^{-1} f_{n+1}$).

To assess whether the sampling method works well, we introduce the exact reconfiguration (ER) that evaluates $S_{i,j}$ and $f$ from $\psi_\theta(x)$ by calculating the exponential sums $\sum_x |\psi_\theta(x)|^2(\cdot)$ exactly, where $x$ is all possible basis vectors in the computational basis (thus we sum over $2^N$ or $\binom{N}{N/2}$ configurations depending on the symmetry of the Hamiltonian).

Within this framework, we classify the difficulty of ground state simulation as follows: We solve the system using the ER with $N = 20$ and the SR with $N = 28$ or 32 (depending

on the symmetry of the Hamiltonian). When the Hamiltonian is free from any of the problems [(i) sampling, (ii) convergence, (iii) expressivity], the converged energies from both methods will be close to the true ground state. If we observe that the ER finds the ground state accurately in a reasonable number of epochs, but SR does not, we conclude that the problem has to do with sampling. In the case of Hamiltonians adiabatically connected to a stoquastic phase, we then change the basis to see whether the sampling problem persists.

If both SR and ER fail, we evaluate the following further diagnostic tests: (a) We compare ER results from several different randomized starting points, and (b) we run the ER through an annealing scheme from a phase that is known to succeed. When all runs of ER return the the same converged energy, we conclude that the problem must be related to expressivity of the ansatz. Otherwise, we try the annealing scheme as an alternative optimization method. If the ER further fails in the annealed scheme, we conclude that the expressivity problem is robust. Finally, we support the classification results from the above procedure by a scaling analysis of the errors for different sizes of the system.

# 3 Preliminary examples

Stoquastic Hamiltonians [18] – those for which all off-diagonal elements in a specific basis are real and non-positive – typically lend themselves to simulation by the path integral quantum Monte Carlo method. However, a "sign problem" arises when this condition is not satisfied, leading to uncontrollable fluctuations of observable quantities as the system grows. The relevance of stoquasticity for the vQMC is far less explored, despite the fact that this method and its variants are often advertized as solving the sign problem [11]. Although it is true that the vQMC is free from summations over alternating signs, the method still has difficulty in solving frustrated Hamiltonians with a complex sign structure as argued in Ref. [16]. In this section, we investigate this question in detail using the one-dimensional Heisenberg XXZ and $J_1$-$J_2$ models the properties of which are well known.

Our strategy is simple. For each Hamiltonian, we use the original Hamiltonian and one with the stoquastic local basis, and observe how the local basis transformation affects the expressivity, convergence, and sampling. Throughout the paper, we will assume periodic boundary conditions for ease of comparison with results from the exact diagonalization.

## 3.1 Heisenberg XXZ and $J_1$-$J_2$ models

The Heisenberg XXZ model is given by

$$H_{\mathrm{XXZ}} = \sum_i \sigma_i^x \sigma_{i+1}^x + \sigma_i^y \sigma_{i+1}^y + \Delta \sigma_i^z \sigma_{i+1}^z, \tag{3}$$

where $\sigma_j^{x,y,z}$ denote the Pauli operators at site $j$, and $\Delta$ is a free (real) parameter of the model. For this Hamiltonian, the Marshall sign rule (applying a Pauli-$Z$ on all even (or odd) sites) changes that Hamiltonian into stoquastic form in the Pauli-$Z$ basis:

$$H'_{\mathrm{XXZ}} = \sum_i -\sigma_i^x \sigma_{i+1}^x - \sigma_i^y \sigma_{i+1}^y + \Delta \sigma_i^z \sigma_{i+1}^z \tag{4}$$

The Hamiltonian is then stoquastic regardless of the value of $\Delta$. Using the RBM with $\alpha = 3$, we plot the result from the ER and SR with and without the sign rule in Fig. 1(a) and (b). For the SR, we sample from the distribution $|\psi_\theta(x)|^2$ using the MCMC with the swap update rule. We mix 16 chains with different temperatures to stabilize sampling.

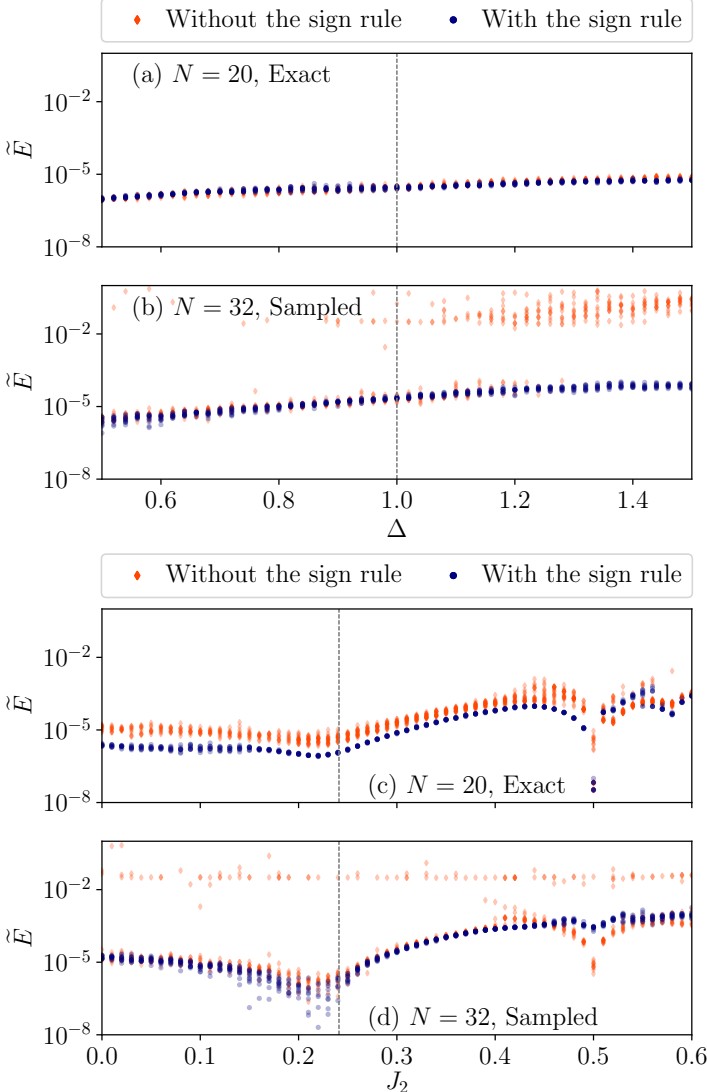

Figure 1:  Converged normalized energy $\widetilde{E} = (E_{\mathrm{RBM}} - E_{\mathrm{ED}})/E_{\mathrm{ED}}$ as a function of model parameters for (a,b) the Heisenberg-XXZ and (c,d) $J_1$-$J_2$ model. For each model, the upper plots [(a) and (c)] presents the results for system size $N = 20$ with the Exact Reconfiguration method that optimizes the parameters using explicitly constructed wavefunctions from the RBM. The lower plots [(b) and (d)] show the results from $N = 32$ with Stochastic Reconfiguration with Markov chain Monte Carlo sampling for optimization. The orange diamonds indicate simulation of the models in the original non-stoquastic basis, while the blue dots indicate simulations in the modified basis, after applying a sigma $Z$ operator on every even sites (the sign rule). Vertical dashed lines indicate where the KT-transitions take place ($\Delta = 1.0$ for the XXZ model and $J_2 \approx 0.2411$ for the $J_1$-$J_2$ model). For each value of the parameters, we have run the simulation 12 times and each point represents the result from a single run.

For each epoch, we use $|\theta| = NM + N + M$ number of samples to estimate $S$ and $f$ (see Sec. 2).

Figure 1(a) clearly shows that the sign rule barely changes the results when we exactly compute the wavefunctions for optimization [2]. This can be attributed to the fact that

---

[2]We note that the converged energies might differ between instances even when the ER is used, due to

the RBM Ansatz can incorporate a Pauli-$Z$ rotation efficiently [28].

On the other hand, when we sample from the distribution [Fig. 1 (b)], some RBM instances fail to find the ground state without the sign rule, especially in the antiferromagnetic phase ($\Delta > 1.0$). Thus we see the *sampling* problem arises due to non-stoquasticity. This is quite surprising, since the MCMC simply uses the ratio between two probability densities $|\psi_\theta(x')/\psi_\theta(x)|^2$, which is sign invariant. However, what happens is more involved. When we use the original Hamiltonian $H_{\text{XXZ}}$, the learning ill-behaves when it hits the regions of parameter space $\theta$ where $S$ and $f$ are not accurately estimated from samples. The transformed Hamiltonian $H'_{\text{XXZ}}$ avoids this problem by following a different learning path. We have observed that in general, the energy of a randomly initialized RBM is much closer to that of the ground state when the sign rule is applied and the learning converges in fewer epochs. We have further tested the SR without the sign rule using different sizes of the system $N = [20, 24, 28, 32]$ and up to $76,800$ samples for each epoch, but observed that the sampling problem persists regardless of such details.

Next, let us consider the one-dimensional $J_1$-$J_2$ model, given by

$$H_{J_1 - J_2} = \sum_i J_1 \boldsymbol{\sigma}_i \cdot \boldsymbol{\sigma}_{i+1} + J_2 \boldsymbol{\sigma}_i \cdot \boldsymbol{\sigma}_{i+2}. \tag{5}$$

where we fix $J_1 = 1.0$. The Hamiltonian has a gapless unique ground state when $J_2 < J_2^*$ (thus, within the critical phase) and gapped two-fold degenerated ground states when $J_2 > J_2^*$. The KT-transition point is approximately known $J_2^* \approx 0.2411$ [29]. In addition, an exact solution at $J_2 = 0.5$ is known – the Majumdar-Ghosh point. The Marshall sign rule also can be applied to this Hamiltonian which yields:

$$H'_{J_1 - J_2} = \sum_i J_1 [-\sigma_i^x \sigma_{i+1}^x - \sigma_i^y \sigma_{i+1}^y + \sigma_i^z \sigma_{i+1}^z]$$
$$+ J_2 \boldsymbol{\sigma}_i \cdot \boldsymbol{\sigma}_{i+2}. \tag{6}$$

We note that this Hamiltonian is still non-stoquastic when $J_2 > 0$.

In Appendix A, we prove that on-site unitary gates that transform $H_{J_1 - J_2}$ into a stoquastic form indeed do not exist when $J_2 > 0$. We also show that ground states in the gapped phase ($J_2 > J_2^*$) cannot be transformed into a positive form easily using the results from Ref. [30].

Simulation results for this Hamiltonian are presented in Fig. 1(c) and (d). First, as in the XXZ model, the ER results in Fig. 1(c) show that the sign rule is not crucial when we exactly compute the observables, i.e. the ER with and without the sign rule both converges to almost the same energies. However, in contrast to the XXZ model, there is a range of $J_2 \in (J_2^*, 0.5) \cup (0.5, 0.6)$ where all ER and SR instances perform badly (the error is $> 10^{-4}$ for some instances) even when the sign rule is applied [Fig. 1(c)]. It indicates that the *expressive power* of the network is insufficient for describing the ground state even though the system is gapped. We further show (see Appendix B) that this problem cannot be overcome by increasing the number of hidden units. Since this region cannot be annealed into from a stoquastic point ($J_2 = 0$) without a phase transition, we argue that this parameter region is in a "deep non-stoquastic" phase.

With the SR, the results in Fig. 1(d) show that the sign rule is crucial in this case. Without the sign rule, some of the instances always fail to converge to true ground states regardless of the Hamiltonian parameters. This is the behavior what we saw from the XXZ model that non-stoquasticity induces a sampling problem. On the other hand, when the sign rule is applied, all SR instances report small relative errors when $J_2 \leq J_2^*$ even though

---

random initialization.

the transformed Hamiltonian is still non-stoquastic. We speculate that this is because the whole region is phase connected to the stoquastic $J_2 = 0$ point.

We summarize the results from the above two models with the following key observations.

**Observation 1.** *Complex RBMs represent ground states of spin chains faithfully when the Hamiltonian is stoquastic, up to a basis transformation consisting of local Pauli gates, or is phase connected to such a Hamiltonian.*

**Observation 2.** *There exists "a deep non-stoquastic phase", where the Hamiltonian cannot be locally or adiabatically transformed into a stoquastic Hamiltonian without crossing a phase transition. Complex RBMs have difficulty representing such ground states.*

**Observation 3.** *Sampling is stable along the learning path when the Hamiltonian is stoquastic or phase connected to a stoquastic Hamiltonian.*

In the next section, we will explore these observations more closely by introducing a more challenging example that combines all of the problems above.

# 4   Further experiments

In previous examples, local basis transformations only marginally affected the expressive power of the model. Here, we introduce a Hamiltonian that requires more than the Pauli gates for a stoquastic transformation. The main findings in this section are (1) local basis transformations beyond the Pauli gates are useful for expressivity, (2) there is a conventional symmetry broken phase for which the RBM fails to represent the ground states, and (3) the number of samples to estimate observables correctly may scale poorly even for a stoquastic Hamiltonian.

## 4.1   Model Hamiltonian and phase diagram

We consider a next nearest neighbor XYZ type Hamiltonian with "twisted" interactions:

$$
\begin{aligned}
H_{\mathrm{tXYZ}} = J_1 \sum_{i=1}^{N} & a\sigma_i^x \sigma_{i+1}^x + b\sigma_i^y \sigma_{i+1}^y + \sigma_i^z \sigma_{i+1}^z \\
+ J_2 \sum_{i=1}^{N} & b\sigma_i^x \sigma_{i+2}^x + a\sigma_i^y \sigma_{i+2}^y + \sigma_i^z \sigma_{i+2}^z
\end{aligned}
\tag{7}
$$

where $a$ and $b$ are two real parameters. Note that $a$ ($b$) is the strength of $XX$ ($YY$) interaction for nearest-neighbors whereas it is on $YY$ ($XX$) interaction for next-nearest-neighbors. This particular Hamiltonian has a rich phase structure as well as stoquastic to non-stoquastic transitions. The stochastic regions do not coincide with the phases of the model. In addition, the system has $\mathbb{Z}_2$ symmetries in any axis $\sigma_i^{\{x,y,z\}} \leftrightarrow -\sigma_i^{\{x,y,z\}}$ as well as translational symmetry. Moreover, a $\pi/2$ rotation of the $z$-axis, i.e. $U = e^{-i\pi/4 \sum_i \sigma_i^z}$, swaps the parameters $a$ and $b$.

Throughout the section, we assume ferromagnetic interactions $J_1 = J_2 = -1$. As the Hamiltonian in this case becomes the classical ferromagnetic Ising model when $a = b = 0$, one may expect that vQMC works well at least for small parameters. However, we will see that this intuition is generally misleading, as the sign structure of the model plays a very

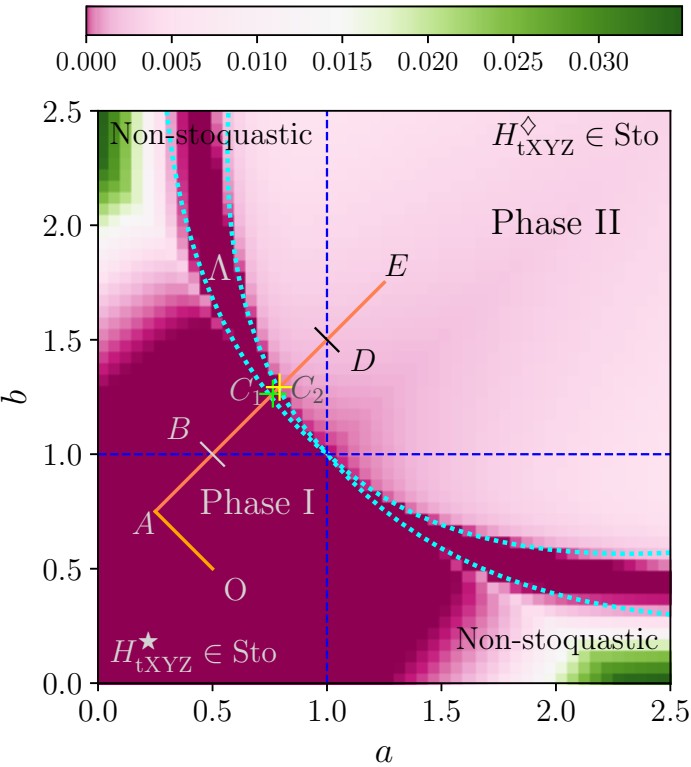

Figure 2: Phase and stoquasticity diagrams of the twisted XYZ model. For parameters $0 \leq a, b \leq 2.5$, the difference between the lowest energies $(E_0 - E_1)/E_0$ in the symmetric and the anti-symmetric subspace under the spin flip $(\sigma_z \leftrightarrow -\sigma_z)$ for $N = 28$ is shown. As the ground states can break the $\mathbb{Z}_2$ symmetry in all three directions, we cannot determine phases solely from this plot. Thus we calculate magnetic susceptibilities in Fig. 3 and find that Phase I breaks the symmetry along the $z$-axis whereas Phase II recovers this. Between those two phases, phase $\Lambda$ that breaks the symmetry along the $y$-axis appears when $a \neq b$. We depict approximate phase boundaries with dotted curves. On the other hand, dashed lines show stoquastic to non-stoquastic transitions. Local basis transformed Hamiltonians $H_{\text{tXYZ}}^{\Diamond}$ and $H_{\text{tXYZ}}^{\star}$ are stoquastic in the first and third quadrants, respectively. In the second and forth quadrants, local (on-site) unitary gates that transforms the Hamiltonian into a stoquastic form do not exist. The line segment from $O = (0.5, 0.5)$ to $A = (0.25, 0.75)$ and $A$ to $E = (1.25, 1.75)$ indicate the parameters we simulated with vQMC. The phase transitions along this line segment take place at $C_1 \approx (0.764, 1.264)$ and $C_2 \approx (0.793, 1.293)$.

important role. We consider two other representations of the model, which are obtained

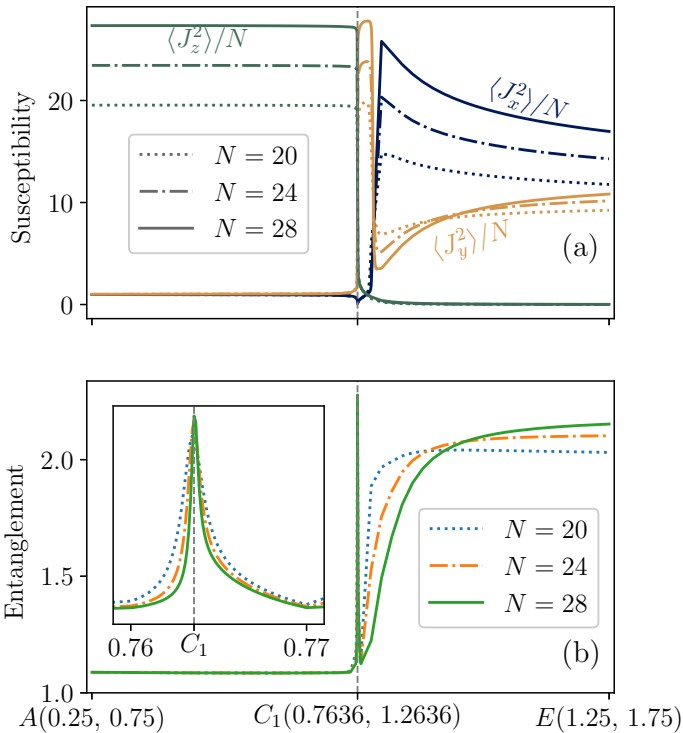

Figure 3: (a) Magnetic susceptibility and (b) entanglement entropy for the ground state of the twisted XYZ model along the line $A = (0.25, 0.75)$ to $E = (1.25, 1.75)$. The result shows that there are three distinct phases. We locate the first phase transition point $C_1 \approx (0.7636, 1.636)$ that shows the divergence of entanglement entropy. The maximum values of the entanglement entropy are 2.16, 2.25, 2.28 for $N = 20, 24$, and 28, corroborating a logarithmic divergence of the entanglement entropy at criticality.

by local basis transformations.

$$H^{\bigstar}_{\text{tXYZ}} = J_1 \sum_{i=1}^{N} \sigma_i^x \sigma_{i+1}^x + b\sigma_i^y \sigma_{i+1}^y + a\sigma_i^z \sigma_{i+1}^z$$

$$+ J_2 \sum_{i=1}^{N} \sigma_i^x \sigma_{i+2}^x + a\sigma_i^y \sigma_{i+2}^y + b\sigma_i^z \sigma_{i+2}^z, \tag{8}$$

$$H^{\diamondsuit}_{\text{tXYZ}} = J_1 \sum_{i=1}^{N} a\sigma_i^x \sigma_{i+1}^x + \sigma_i^y \sigma_{i+1}^y + b\sigma_i^z \sigma_{i+1}^z$$

$$+ J_2 \sum_{i=1}^{N} b\sigma_i^x \sigma_{i+2}^x + \sigma_i^y \sigma_{i+2}^y + a\sigma_i^z \sigma_{i+2}^z. \tag{9}$$

The Hamiltonian $H^{\bigstar}_{\text{tXYZ}}$ ($H^{\diamondsuit}_{\text{tXYZ}}$) is stoquastic for $0 \leq a, b \leq 1$ ($a, b \geq 1$), and can be obtained by applying $\pi/2$ rotation over $y$ ($x$) axes, respectively. We note that those rotations involve the Hadamard gate, so they cannot be decomposed only by Pauli gates, e.g. $\pi/2$ rotation over the $y$-axis is given by $e^{-i\pi/4Y} = XH$. These Hamiltonians are obtained by applying the general local transformations described by Klassen and Terhal [31] to Eq. (7) (see Appendix C for detailed steps).

Before presenting vQMC results, let us briefly summarize the phase structure of the Hamiltonian that is presented in Fig. 2. To gain an insight, let us first consider the $a = b$ line. When $a = b < 1.0$, each term of the Hamiltonian prefers an alignment in $z$-direction so the ground state is $|\uparrow\rangle^{\otimes N} + |\downarrow\rangle^{\otimes N}$. Even though the $U(1)$ symmetry is broken when $a \neq b$, this ferromagnetic phase extends from $a = b < 1.0$ which we denote by Phase I in Fig. 2. On the other hand, the Hamiltonian prefers the total magnetization $J_z = \sum_i \sigma_i^z = 0$ when $a = b > 1.0$. The region of this phase is shown in Fig. 2 denoted by Phase II. As the total magnetization changes abruptly at $a = b = 1$ regardless of the system size, we expect a first order phase transition to take place at this point. However, the phase boundaries when $a \neq b$ are more complex and another phase $\Lambda$ appears in between two phases.

To characterize the phases when $a \neq b$, we plot magnetic susceptibilities and the entanglement entropy along the line segment $\overline{AE}$ in Fig. 3. For each parameter $(a, b)$, we have obtained the ground state within the subspace preserving the $\mathbb{Z}_2$ symmetry along the $z$-axis using exact diagonalization (thus our ground states obey the $\mathbb{Z}_2$ symmetry even when the symmetry is broken in the thermodynamic limit). We see that the magnetic susceptibility along the $z$-axis diverges as $N$ increases at Phase I which implies the symmetry will be broken when $N \to \infty$. Likewise, we also see that the symmetry along the $y$ axis is broken at Phase $\Lambda$. We see that entanglement entropy diverges at $C_1 \approx (0.764, 1.264)$ along the $AE$ line, suggesting a second order phase transition.

However, the signature of the second phase transition from the entanglement entropy is weak possibly because the phase transition is the infinite order Kosterlitz–Thouless transition. Thus we calculate the second derivative of the ground state energy in Appendix D and locate the second phase transition point $C_2 \approx (0.793, 1.293)$.

In addition, entanglement entropy shows that there is no hidden order in Phase I and $\Lambda$ as it is near to 1.0 which can be fully explained by the broken $\mathbb{Z}_2$ symmetry. We also see a signature of other phases when $a$ is small and $b$ is large (or vice versa). Though, we will overlook them as they are far from the parameter path we are interested in.

We note that even though the phases in Fig. 2 are identified following the conventional $\mathbb{Z}_2$ symmetry breaking theory, we will further restrict a symmetry class of the Hamiltonian when we discuss phase connectivity throughout the section as it provides a more consistent view. For example, we will consider that point $O$ (located on $a = b$ line which has the $U(1)$ symmetry) is not phase connected to $A$ (where the Hamiltonian obviously breaks the $U(1)$ symmetry), whereas $A$ and $B$ are phase connected. Our definition of phase connectivity is also compatible with a modern definition of phases in a one-dimensional system [32–34].

Finally, we depict the regions of stoquasticity in Fig 2. The model can be made stoquastic by a local basis change in the bottom left and top right quadrants. Within this phase diagram, we run our vQMC simulations along two paths $\overline{OA}$ and $\overline{AE}$. The path $\overline{OA}$ does not cross any phase or stoquasticity boundary, but it will show how a symmetry of the ground state affects the expressivity. On the other hand, the path $\overline{AE}$ crosses both the phase and stoquasticity boundaries thus will show how phase and stoquasticity transitions affect vQMC.

## 4.2  Variational Quantum Monte Carlo results

Our vQMC results for the twisted nnn. XYZ Hamiltonian are shown in Fig. 4. The shades in the middle of Fig. 4(c) and (d) indicate the region where the model cannot be made stoquastic by a local rotation. We discuss the results for each path and phase below. As the phase $\Lambda$ (located between $C_1$ and $C_2$) is disconnected from others, we examine this case separately at the end of the section.

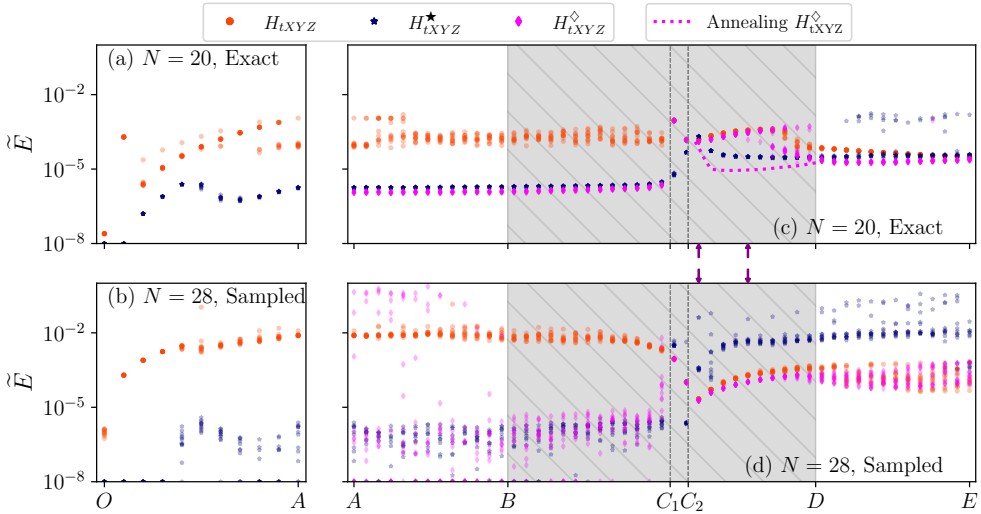

Figure 4: Normalized energy from the vQMC for the twisted XYZ model. We have used the ER with $N = 20$ for (a) and (c), the SR with $N = 28$ for (b) and (d). (a,b) For $\overline{OA}$, the original Hamiltonian $H_{\mathrm{tXYZ}}$ is non-stoquastic whereas $H_{\mathrm{tXYZ}}^{\star}$ is stoquastic. (c,d) Hamiltonians $H_{\mathrm{tXYZ}}^{\star}$ and $H_{\mathrm{tXYZ}}^{\Diamond}$ are stoquastic on the left and right side of the shaded region, respectively. In the shaded region, none of the Hamiltonians is stoquastic. The original Hamiltonian $H_{\mathrm{tXYZ}}$ is non-stoquastic for all parameters. Vertical dashed lines at $C_1$ and $C_2$ indicate the phase transition points. A dotted curve in (c) indicates an annealing result (see main text).

### 4.2.1  Path $\overline{OA}$ (Phase I)

As we have noted above, the ground state is a classical ferromagnet at location $O$. We observe that the RBM represents this state as expected even when we use the original Hamiltonian $H_{\mathrm{tXYZ}}$ which is non-stoquastic at this point. However, the error from the ER is getting larger as the parameter deviates from $O$. Since the Hamiltonian is always gapped along $\overline{OA}$, this result shows that non-stoquasticity affects vQMC even though the path does not close a gap; thus it reveals the importance of symmetry in the RBM expressivity.

With our symmetry sensitive definition of phase connectivity, the solubility indeed can be understood well as follows. First, the ground state at $O$ is represented by the RBM using the original Hamiltonian $H_{\mathrm{tXYZ}}$ as it is phase connected to the $a = b = 0$ point (Observation 1). However, as going to $A$ breaks the $U(1)$ symmetry and it cannot be transformed into a stoquastic form only using local Pauli gates (see Appendix C), point $A$ is not guaranteed to be solvable using $H_{\mathrm{tXYZ}}$. On the other hand, one can solve it using $H_{tXYZ}^{\star}$ which is stoquastic along the whole path $\overline{OA}$.

The fact that the transformed Hamiltonian $H_{tXYZ}^{\star}$ works much better than the original one $H_{tXYZ}$ in the ER case contrasts to the XXZ and $J_1$-$J_2$ models where local Pauli rotations affected the expressive power moderately. This is because the transformations in this model involve the Hadamard gate which is known to be challenging for the RBM [28, 35].

#### 4.2.2   Path $\overline{AC_1}$ (Phase I)

We observe that the transformed Hamiltonians $H_{\text{tXYZ}}^{\bigstar}$ and $H_{\text{tXYZ}}^{\Diamond}$ both work better than the original Hamiltonian $H_{\text{tXYZ}}$ along this path when the ER is used [Fig. 4(c)]. This fact reinforces Observation 1. Moreover, the results in the shaded region $(\overline{BC_1})$ where none of the Hamiltonians is stoquastic, display almost the same errors as the region $(\overline{AB})$. This confirms that phase transitions affects expressivity much more than stoquastic to non-stoquasticity transitions (Observation 1 and 2).

In contrast, the results from the SR [Fig. 4(d)] show that $H_{\text{tXYZ}}^{\bigstar}$ which is stoquastic on the left side of the shaded region works better than $H_{\text{tXYZ}}^{\Diamond}$. This result indicates that the MCMC is more stable when the stoquastic Hamiltonian is used, which is the behavior we have seen from the sign rule of the XXZ and the $J_1$-$J_2$ models (Obervation 3). Interestingly, $H_{tXYZ}^{\Diamond}$ appears to be sensitive to the stoquastic transition although it is non-stoquastic throughout the path. We do not have a good explanation for this behavior.

#### 4.2.3   Path $\overline{C_2D}$ (Phase II)

We observe that $H_{\text{tYXZ}}^{\Diamond}$ which is stoquastic to the right of $D$ does not give the best result in this region $\overline{C_2D}$ when the ER is used. However, a large fluctuation in the converged energies suggests that the *convergence* problem [case (ii)] arises, likely due to a complex optimization landscape. Thus we need to distinguish the problem between optimization and expressivity more carefully in this region.

For this purpose, we use an annealing approach as an alternative optimization method. For annealing, we first take converged RBM weights when $(a, b) = (1.01, 1.51)$ (the point right next to $D$) where the Hamiltonian $H_{\text{tYXZ}}^{\Diamond}$ is stoquastic. Then we run the ER from these weights instead of randomly initialized ones. We decrease the parameters $(a, b)$ of the Hamiltonian by $(0.01, 0.01)$ for each annealing step and run 200 ER epochs. The obtained results are indicated by a dotted curve in Fig. 4(c). The annealing result suggests that the expressivity is not the main problem up to the phase transition point $C_2$ (when considered from the right to the left).

The SR results show two noteworthy features compared to the ER results. First, the Hamiltonian $H_{\text{tXYZ}}^{\bigstar}$ gives remarkably poor converged energies compared to the results from the ER. This result agrees with what we have seen from the sign rule: When the Hamiltonian is non-stoquastic, the learning path may enter a region where the sampling becomes unstable. Second, the shape of the curves from the Hamiltonians $H_{\text{tXYZ}}$ and $H_{\text{tXYZ}}^{\Diamond}$ are also different from that of the ER which is due to poor optimization. However, in Appendix E, we show that the *convergence* problem gets weaker as $N$ increases, thus the SR can solve the Hamiltonian $H_{\text{tXYZ}}^{\Diamond}$ in this region correctly for a large $N$ by examining the two parameter points of the Hamiltonian (indicated by arrows in Fig. 4).

We encapsulate the results in this region as follows: The vQMC works for $H_{\text{tYXZ}}^{\Diamond}$ that is phase connected to a stoquastic Hamiltonian even though it suffers from a optimization problem for small $N$. This result is consistent with Observation 1 and Observation 3.

#### 4.2.4   Path $\overline{DE}$ (Phase II)

The ER results show that $H_{\text{tXYZ}}$ and $H_{\text{tYXZ}}^{\Diamond}$ both work better than $H_{\text{tXYZ}}^{\bigstar}$. It is interesting that the non-transformed one $H_{\text{tXYZ}}$ works in this region despite its non-stoquasticity. However, the SR results show that the converged energies from $H_{\text{tXYZ}}$ and $H_{\text{tXYZ}}^{\Diamond}$ suffer large fluctuations, which suggests that the sampling induces errors. This result is unexpected as the Hamiltonian $H_{\text{tXYZ}}^{\Diamond}$ is stoquastic in this region.

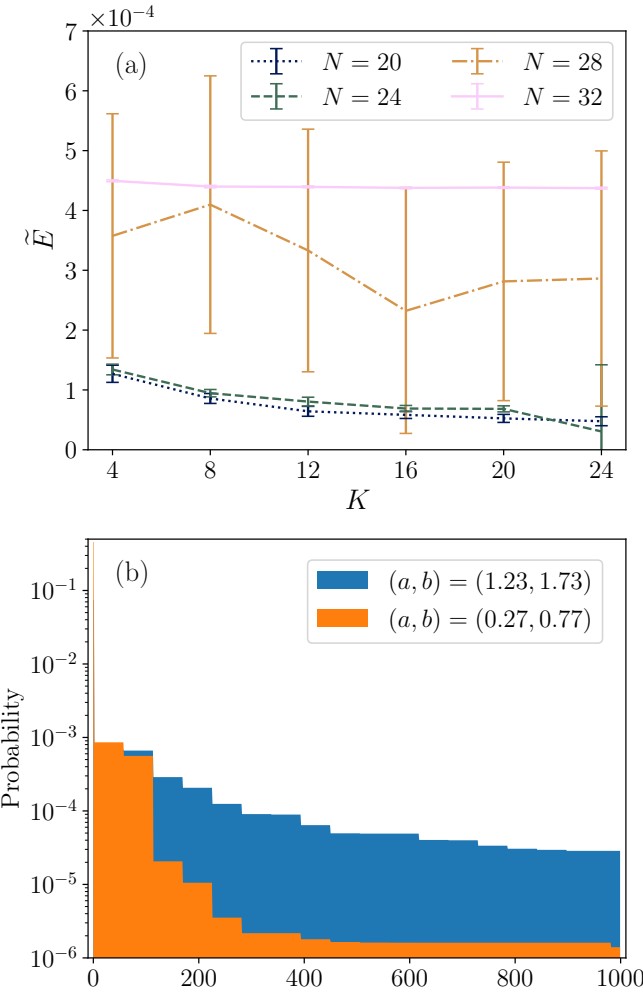

Figure 5: (a) Normalized energy from the vQMC after convergence as a function of number of samples for different system sizes $N$. The Hamiltonian $H_{\text{tXYZ}}^{\diamond}$ with $(a, b) = (1.23, 1.73)$ is simulated. We have used $K \times |\theta|$ number of samples to estimate observables in each update step of the SR. The result is averaged over 12 vQMC instances and error bars indicate the standard deviation. Error bars for $N = 32$ are invisible as they are less than $2.0 \times 10^{-6}$. The result shows there is a transition in scaling near $N = 28$. (b) The first $10^3$ elements of $|\psi_{\text{gs}}(x)|^2$ where $\psi_{\text{gs}}(x)$ is the ground state of $H_{\text{tXYZ}}^{\diamond}$ obtained from the exact diagonalization when $N = 28$. Parameters $(a, b) = (0.27, 0.77)$ in Phase I and $(1.23, 1.73)$ in Phase II are used. When $(a, b) = (0.27, 0.77)$, the peak at the beginning indicate two largest elements of the distribution which are $\approx 0.458$. We see that the tail distribution for Phase II is much thicker. Moreover, the summation of the first $10^3$ elements is $\approx 0.998$ when $(a, b) = (0.27, 0.77)$ whereas it is only $\approx 0.294$ when $(a, b) = (1.23, 1.73)$. It suggest that one needs a huge number of samples to correctly estimate the probability distribution in Phase II.

As the Hamiltonian is stoquastic, one may expect that using more samples easily overcomes the problem. However, this is not the case. To see this, we plot vQMC errors as a function of the number of samples in Fig. 5(a). Here, the $K$ in $x$-axis means that we have used $K \times |\theta|$ samples per epoch. For $N = 20$ and $24$, one observes that the errors get smaller as the number of samples increase. However, the results are subject to large fluctuations at $N = 28$, and gets worse at $N = 32$. Since $\theta$ itself scales as $\sim \alpha N^2$, our

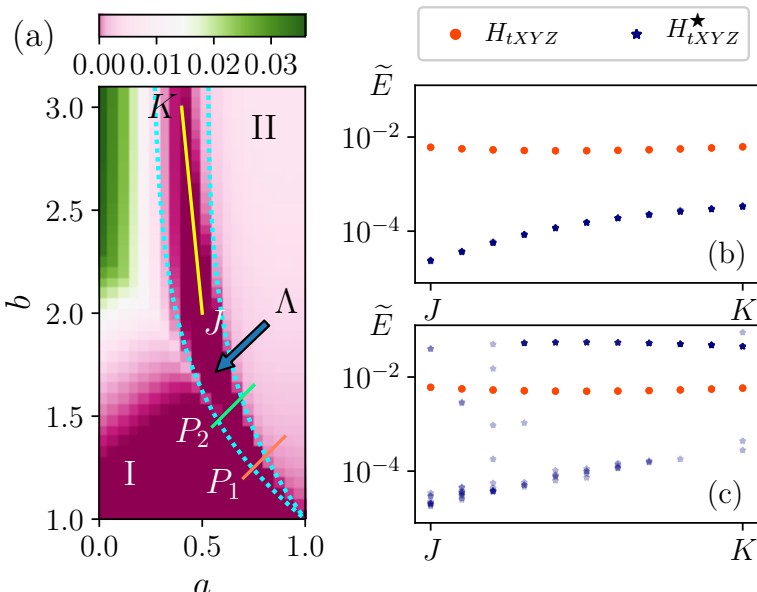

Figure 6: (a) The second quadrant of the phase diagram in Fig. 2. We calculate the second derivative of the ground state energy along paths $P_1$ and $P_2$ in Appendix D to locate the phase transition points. Converged normalized energy using (b) the ER with $N = 20$ and (c) the SR with $N = 28$ along the path $\overline{JK}$ where $J = (0.5, 2.0)$ and $K = (0.4, 3.0)$.

results show that this sampling problem is robust.

It is insightful to compare the sampling problem in this model to that in non-stoquastic models (such as the XXZ model in the antiferromagnetic phase without the sign rule). Even though they are both caused by a finite number of samples, the converged energies suggest that they have different origins. In the XXZ model, the converged normalized energies are mostly clustered above $10^{-2}$ regardless of the size of the system. On the other hand, they are below $10^{-3}$ and show clear system size dependency in this model. In Appendix F, we show that the sampling problem in this model only appears locally near the minima whereas it pops up in the middle of optimizations and spoils the whole learning procedure in non-stoquastic models.

The sampling problem occurring here is quite similar to the problem observed from quantum chemical Hamiltonians [23]. Precisely, Ref. [23] showed that optimizing the RBM below the Hartree-Fock energy for quantum chemical Hamiltonians requires a correct estimation of the tail distribution. However, the tail distribution of the ground state is often thick and a large number of samples are required to estimate observables correctly. We find that the sampling problem of our model is also caused by such a heavy tail in the distribution. We can see this from the probability distribution of the ground states $|\psi_{\mathrm{gs}}(x)|^2$ for $H_{\mathrm{tXYZ}}^{\Diamond}$ using two different parameters $(a, b) = (0.27, 0.77)$ and $(1.23, 1.73)$ which are deep in Phase I and II, respectively. We plot the first $10^3$ largest elements of $|\psi_{\mathrm{gs}}(x)|^2$ for each parameter of the Hamiltonian in Fig. 5(b). The Figure directly illustrates that the distribution of the ground state in Phase II is much broader than that of Phase I. Moreover, the sum of the first $10^3$ elements is only $\approx 0.294$ in Phase II, which implies that one needs a huge number of samples to correctly estimate the probability distribution.

### 4.2.5 Phase Λ

Finally, we show that the RBM has difficulty representing the ground states in phase Λ even though it is a simple conventionally ordered phase (we have observed this from the entanglement entropy). We simulate vQMC along the line $\overline{JK}$ in Fig. 6(a) which is deep in phase Λ. The transformed Hamiltonian $H_{\text{tXYZ}}^{\diamond}$ gives almost the same converged energies as the original Hamiltonian $H_{\text{tXYZ}}$, so we do not present them in Fig. 6(b) and (c). The ER results in Fig. 6(b) clearly show that the error increases as we go deeper in this phase. As in the $J_1$-$J_2$ model, we simulate the ER with varying $N$ and different numbers of hidden units at point $K$ in Appendix G which confirms that there is the *expressivity* problem in this phase. In addition, the SR results [Fig. 6(c)] show that the sampling problem also arises when we use $H_{\text{tXYZ}}^{\star}$ which performed best with the ER. We also note that we cannot apply the results in Ref. [30] as in the $J_1$-$J_2$ model, since there is no hidden orders in this phase.

To summarize overall the results from the twisted XYZ model, we have found that Observation 1 and 2 hold in general by examining the behavior of the vQMC in different phases and stoquastic/non-stoquastic regions. In addition, we also have observed that a different type of sampling problem may arise even when solving a stoquastic Hamiltonian. Thus we modify our observation 3 slightly as follows:

**Observation 3** (Second version). *Sampling is stable along the learning path when the Hamiltonian is stoquastic or phase connected to a stoquastic Hamiltonian. However, the number of samples required to correctly estimate observables may scale poorly even when the Hamiltonian is stoquastic.*

## 5  Remarks

By way of example, we have classified the failure mechanism of the RBM Ansatz within the vQMC framework for one dimensional spin systems. In particular, we revealed how non-stoquasticity affects the expressivity and the MCMC sampling efficiency in vQMC simulations. Most importantly, we provide strong evidence suggesting that the RBM ansatz can faithfully express the ground state of a non-stoquastic Hamiltonian when it is phase connected to a stoquastic Hamiltonian. This observation implies that it may be possible to solve a large number of non-stoquastic Hamiltonians using the RBM Ansatz, significantly expanding the reach of vQMC. On the other hand, our results also suggests that a fundamental difficulty may exist when solving a Hamiltoninan within a phase that is separated from any stoquastic Hamiltonian. Possible examples that involve such a difficulty include a highly frustrated system and topological phases [36, 37].

However, there is a caveat on our "deep non-stoquastic phases" as they rely rather on a conventional concept of phases (phase connectivity is restricted within a given *parameterized* Hamiltonian). In contrast, a modern language of one-dimensional phases allows any constant-depth local unitary transformations that preserve a symmetry between ground states, i.e. two Hamiltonians with the ground states $|\psi_1\rangle$ and $|\psi_2\rangle$ are in the same phase if there is a symmetry preserving unitary operator $U_{\text{sym}}$ such that $|\psi_1\rangle = U_{\text{sym}} |\psi_2\rangle$ and can be decomposed into a constant depth circuit consists of local gates [32–34]. One may possibly interpret our results using symmetry protected phases. Recently, Ref. [30] reported results on a related problem when a ground state can be transformed into a positive form using a unitary operator with a certain symmetry. However, the result there is limited to ground states with hidden orders, while our results suggest that ground states in phase Λ of the twisted XYZ model suffer from the sign problem even though it is conventionally

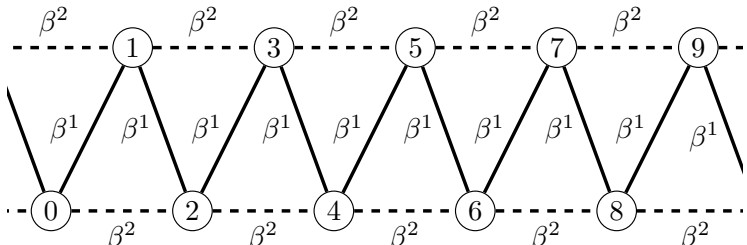

Figure 7:   Graph describing interactions for translational invariant models with the nearest-neighbor and the next-nearest-neighbor couplings.

ordered. Thus, a further study is required to understand how phases of a many-body system interplay with a stoquasticity more deeply.

We also have found that the number of samples to solve the ground state may scale poorly even when the system is stoquastic. This type of difficulty is known from quantum chemical systems [23] but has not been discussed in solving many-body Hamiltonians. We still do not exclude the possibility that an efficient sampling algorithm that correctly estimates observables within a polynomial time may exist even in this case.

Finally, we contrast our results to those of Ref. [17] where a failing mechanism in a supervised learning set-up is investigated. Using signs of the ground state obtained from the exact diagonalization as an input dataset, Ref. [17] showed that a neural network trained using a small subset of $\binom{N}{N/2}$ configurations does not output correct sign information for the remaining set of inputs and concluded that the main difficulty of a frustrated system is sampling rather than expressivity. Even though we have seen a similar type of sampling problem appearing in the RBM (e.g. along path $\overline{DE}$ of the twisted XYZ model the ER converges while the SR fails), we also have found that the expressivity of the network is insufficient in deep non-stoquastic phases by using the ER, which is free from the sampling problem by construction.

*Note added.–* A recent preprint has claimed that *convergence* [case (ii)] is the main problem for solving a non-stoquastic Hamiltonian using neural network Ansatz  [38]. However, they used a different network to ours and only considered a single point in the two dimensional $J_1$-$J_2$ model, hence not capturing the full array of failing mechanisms for neural network states.

## Acknowledgement

The authors thank Prof. Simon Trebst and Dr. Ciarán Hickey for helpful discussions. This project was funded by the Deutsche Forschungsgemeinschaft under Germany's Excellence Strategy - Cluster of Excellence Matter and Light for Quantum Computing (ML4Q) EXC 2004/1 - 390534769 and within the CRC network TR 183 (project grant 277101999) as part of project B01. The numerical simulations were performed on the JUWELS cluster at the Forschungszentrum Juelich. This work presented in the manuscript was completed while both authors we at the University of Cologne.

## A   Non-stoquasticity of the $J_1$-$J_2$ model

In this Appendix, we present an algorithm introduced by Klassen and Terhal [31] that determines (non-)stoquasticity of spin chains, and apply it to the $J_1$-$J_2$ model studied in

the main text.

First, we interpret the XYZ type Hamiltonian as a graph with matrix-valued edges. For each Hamiltonian term between vertices $(i,j)$ given by $H_{i,j} = \beta_{ij}^x \sigma_i^x \sigma_j^x + \beta_{ij}^y \sigma_i^y \sigma_j^y + \beta_{ij}^z \sigma_i^z \sigma_j^z$ we assign a matrix $\beta_{ij} = \mathrm{diag}(\beta_{ij}^x, \beta_{ij}^y, \beta_{ij}^z)$ to edge $(i,j)$. For a many-body spin-1/2 Hamiltonian consisting of two-site terms, it is known that each term much be stoquastic to make the Hamiltonian stoquastic. Using the matrix $\beta$, the term $H_{i,j}$ is stoquastic when $\beta_{ij}^x \leq -|\beta_{ij}^y|$. In addition, if there is a local (on-site) basis transformation that converts the Hamiltonian into a stoquastic form, it must act as signed permutations to each vertex, i.e. transform the matrix $\beta_{ij}$ to $\widetilde{\beta}_{ij} = \Pi_i \beta_{ij} \Pi_j^T$ where $\Pi_i = R_i \Pi_i$, $R_i = \mathrm{diag}(\pm 1, \pm 1, \pm 1)$ is a sign matrix and $\Pi_i \in S_3$ is a permutation (Lemma 22 of Ref. [31]). One may note that for signed permutations, the transformed Hamiltonian is still XYZ-type ($\widetilde{\beta}_{ij}$ are diagonal). Then we can formally write the problem as: find a set of signed permutations $\{\Pi_i = R_i \Pi_i\}$ that makes the transformed matrix $\widetilde{\beta}_{ij} = \Pi_i \beta_{ij} \Pi_j^T$ satisfy $\widetilde{\beta}_{ij}^x \leq -|\widetilde{\beta}_{ij}^y|$ for all edges $(i,j)$. The algorithm solves this by separating the permutation and the sign parts.

1. First, check whether there is a set of permutations $\{\Pi_i \in S_3\}$ that make $|\widetilde{\beta}_{ij}^x| \geq |\widetilde{\beta}_{ij}^y|$. This step can be done efficiently utilizing the fact that permutations $\Pi_i$ and $\Pi_j$ must be the same if the rank of $\beta_{ij}$ is $\geq 2$ (Lemma 23 of Ref. [31]).

2. Second, for possible permutations obtained from the above step, check if there is a possible set of signs $\{R_i\}$ that makes $\widetilde{\beta}_{ij}^x$ negative. This is reduced to a problem of deciding whether to apply $\mathrm{diag}(-1, 1, 1)$ or not to each vertex (Lemma 27 of Ref. [31]), which is equivalent to solving the frustration condition of classical Ising models that can be solved in a polynomial time.

For detailed steps, we refer to the original work in Ref [31]. We also note that the algorithm finds a solution *if and only if* such a transformation exists.

We depict the interaction graph for translational invariant models with the next-nearest-neighbor couplings in Fig. 7. For this type of graph, the second problem (finding the possible signs) is much easier to solve. In our graph, applying $\mathrm{diag}(-1, 1, 1)$ to all even sites flips the sign of $\beta_1^x$. However, such a sign rule does not exist for $\beta_2^x$. Thus after the permutation, $\widetilde{\beta}_2^x$ must be negative whereas the sign of $\widetilde{\beta}_1^x$ for the nearest-neighbor couplings is free as we can always make it negative by applying this sign rule.

We can directly apply this principle to the $J_1$-$J_2$ model. For the $J_1$-$J_2$ model we have studied in the main text, we have $\beta$ matrices:

$$\beta_1 = J_1 \begin{pmatrix} 1 & 0 & 0 \\ 0 & 1 & 0 \\ 0 & 0 & 1 \end{pmatrix}, \qquad \beta_2 = J_2 \begin{pmatrix} 1 & 0 & 0 \\ 0 & 1 & 0 \\ 0 & 0 & 1 \end{pmatrix}. \tag{10}$$

As a permutation does not change the matrices, we only need to consider the sign rule which readily implies the Hamiltonian is stoquastic only when $J_2 \leq 0$. Still, it is known that the $J_1$-$J_2$ model can be transformed into a stoquastic form using two-qubit operations [39].

In addition to the non-stoquasticity of the Hamiltonian, one can further prove that ground states in the gapped phase of this model (when $J_2 > J_2^*$) indeed cannot be transformed into a positive form easily. Reference [30] states that a phase that is short-ranged entangled and has a string order suffers a symmetry protected sign problem – the transformed ground state in the computational basis cannot be positive (there exists $x$ such that $\langle x | U_{\mathrm{sym}} | \psi_{\mathrm{GS}} \rangle \notin \mathbb{R}_{\geq 0}$) for all symmetry protecting unitary operators $U_{\mathrm{sym}}$. We can directly apply this result to our case as the phase of the $J_1$-$J_2$ model when $J_2 > J_2^*$ satisfies this condition (see e.g. Ref. [40]). As the symmetric group is $G = SO(3)$ in the

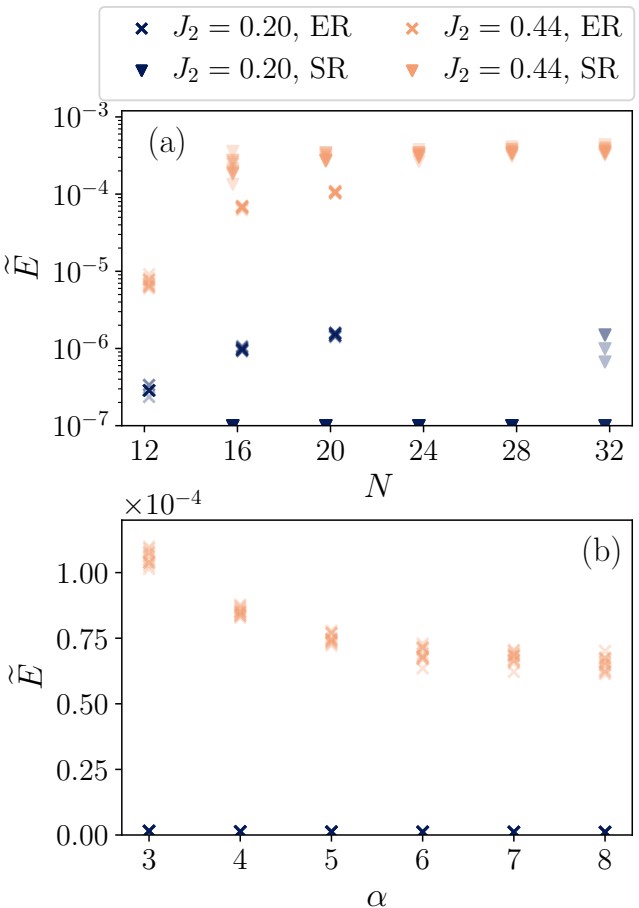

Figure 8: (a) Converged normalized energies $\widetilde{E} = (E_{\mathrm{RBM}} - E_{\mathrm{ED}})/E_{\mathrm{ED}}$ from the ER and the SR as a function of different system size $N$ for the J₁-J₂ model with $J_2 = 0.2$ and $0.44$. We applied the sign rule and the ratio between hidden and visible units $\alpha = 3$ is used. (b) Converged normalized energies as a function of the number of hidden units $\alpha = M/N$ when $N = 20$.

J₁-J₂ model, this implies that a translational invariant gate $U^{\otimes N}$ which transforms the ground state into a positive form does not exist. This further supports a relation between the expressive power of the RBM and stoquasticity of the Hamiltonian. Interestingly, at the Majumdar-Ghosh point ($J_2 = 0.5$), a non-translational invariant unitary gate $\bigotimes_i \sigma^z_{2i}$ transforms the ground state into a positive form yet the transformed Hamiltonian is still non-stoquastic.

## B  Expressive power of the RBM for the J₁-J₂ model

In the main text, we have observed that the errors from the J₁-J₂ model when $J_2 \in (J_2^*, 0.5)$ are large even when the ER is used and the sign rule is applied. In this section, we investigate the errors in more detail using the RBM with different system sizes and number of hidden units.

For $J_2 = 0.20$ and $0.44$, we plot converged normalized energies for different system sizes $N$ and values of $\alpha = M/N$ in Fig. 8. Figure 8(a) shows that both the ER and the

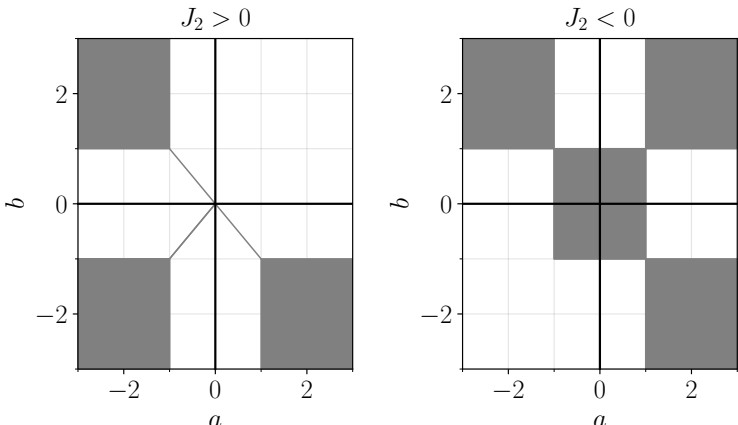

Figure 9: Parameter regions that the twisted XYZ model can be transformed into a stoquastic form by on-site unitary gates.

SR find the ground state when $J_2 = 0.2$. We also observe the MCMC samples bias the energy a little toward the ground state and the energy and the statistical fluctuation get larger as $N$ increases in the SR case. However, one can easily deal with such a problem by carefully choosing the sample size. In contrast, we see that the expressivity problem arises when $J_2 = 0.44$, which is more fundamental. In addition, even though the errors seem to remain constant with $N$ for the SR, we expect the obtained state to be moving away from the true ground state as $N$ increases because the normalized energy of the first excited state scales as $\sim 1/N$.

We additionally study how the error scales with the number of hidden units in Fig. 8(b). The results clearly show that increasing $\alpha = M/N$ only marginally reduces the error. We also note that the system at $J_2 = 0.44$ is gapped thus the error from $\alpha = 8$ is still far from the error we expect from other methods e.g. density matrix renormalization group. This supports our argument that the RBM has a difficulty in representing the ground state of Hamiltonians that is deep in non-stoquastic phase. In addition, Ref. [41] has reported similar behavior of errors for one dimensional $J_1$-$J_2$ model using long-range entangled plaquette states. This suggests that our observations for *expressivity* can be universal among many classically computable Ansatz states.

## C    Non-stoquasticity of the twisted XYZ model

In Appendix A, we introduced the algorithm by Klassen and Terhal and applied it to the $J_1$-$J_2$ model. In this section, we apply the algorithm to the twisted XYZ model and determine the parameter regions where the twisted XYZ model can be locally transformed into stoquastic form.

For the twisted XYZ model, the $\beta$ matrices are given as

$$\beta_1 = J_1 \begin{pmatrix} a & 0 & 0 \\ 0 & b & 0 \\ 0 & 0 & 1 \end{pmatrix}, \qquad \beta_2 = J_2 \begin{pmatrix} b & 0 & 0 \\ 0 & a & 0 \\ 0 & 0 & 1 \end{pmatrix}. \tag{11}$$

The model is trivially stoquastic when $a = b = 0$. If any of $a$ or $b$ is non-zero, we see the ranks of the matrices are $\geq 2$. Thus all permutations acting on each vertex must be equal, i.e. $\Pi_i = \Pi$. After this simplification, we can just apply each element $\Pi \in S_3$ and see whether there is a set of sign matrices that satisfies the condition.

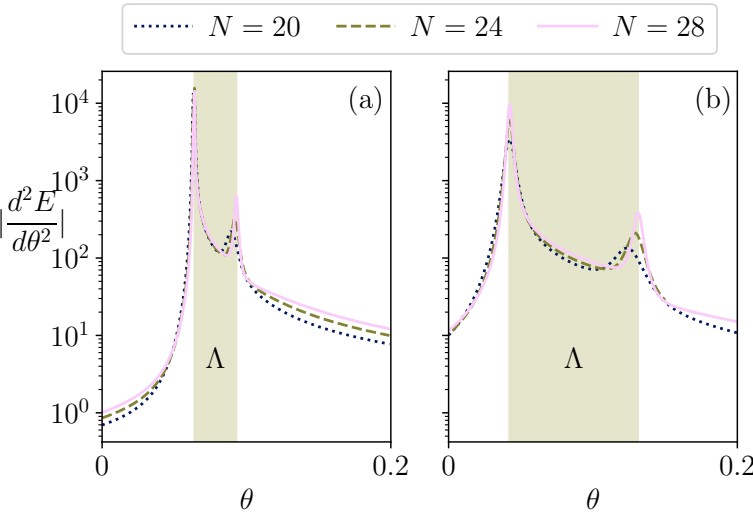

Figure 10: Second derivatives of the ground state energies along two paths (a) $P_1$ and (b) $P_2$ depicted in Fig. 6 of the main text. Path $P_1$ is from $(a, b) = (0.7, 1.2)$ to $(0.9, 1.4)$ that is a part of $\overline{AE}$ in Fig. 2 that we simulated vQMC. Path $P_2$ is from $(0.55, 1.45)$ to $(0.75, 1.65)$.

For our model, it is not difficult to obtain the conditions below:

$$\Pi = () \qquad |a| = |b| \cap J_2 b \leq 0 \tag{12}$$

$$\Pi = (1, 2) \qquad |b| = |a| \cap J_2 a \leq 0 \tag{13}$$

$$\Pi = (2, 3) \qquad |a| \geq 1 \cap |b| \geq 1 \cap J_2 b \leq 0 \tag{14}$$

$$\Pi = (1, 3) \qquad 1 \geq |b| \cap 1 \geq |a| \cap J_2 \leq 0 \tag{15}$$

$$\Pi = (1, 2, 3) \qquad 1 \geq |a| \cap 1 \geq |b| \cap J_2 \leq 0 \tag{16}$$

$$\Pi = (1, 3, 2) \qquad |b| \geq 1 \cap |a| \geq 1 \cap J_2 a \leq 0 \tag{17}$$

which is depicted in Fig. 9.

When $J_1 = J_2 = -1$ and $a, b \geq 0$, that we considered in the main text, we see that $\Pi = (2, 3)$ and $\Pi = (1, 3)$ are the transformations that make the Hamiltonian stoquastic for $a, b \geq 1$ and $a, b \leq 1$, respectively. As the elements of the transformed matrices $\widetilde{\beta}_1 = \Pi \beta_1 \Pi^T, \widetilde{\beta}_2 = \Pi \beta_2 \Pi^T$ are already negative, additional application of the sign rule is not required.

## D    Phase transitions in the twisted XYZ model

In the main text, we summarized the phases of the twisted XYZ model. Here, we investigate the phases more closely and also locate the second phase transition point along the path $\overline{AE}$. Let us first consider the case when $a = b$ that recovers the XXZ model with the next-nearest-neighbor couplings. After setting $J_1 = J_2 = -1$, the Hamiltonian becomes

$$H_{tXYZ} = \sum_i -(C\sigma_i^x \sigma_{i+1}^x + C\sigma_i^y \sigma_{i+1}^y + \sigma_i^z \sigma_{i+1}^z)$$
$$- (C\sigma_i^x \sigma_{i+2}^x + C\sigma_i^y \sigma_{i+2}^y + \sigma_i^z \sigma_{i+2}^z) \tag{18}$$

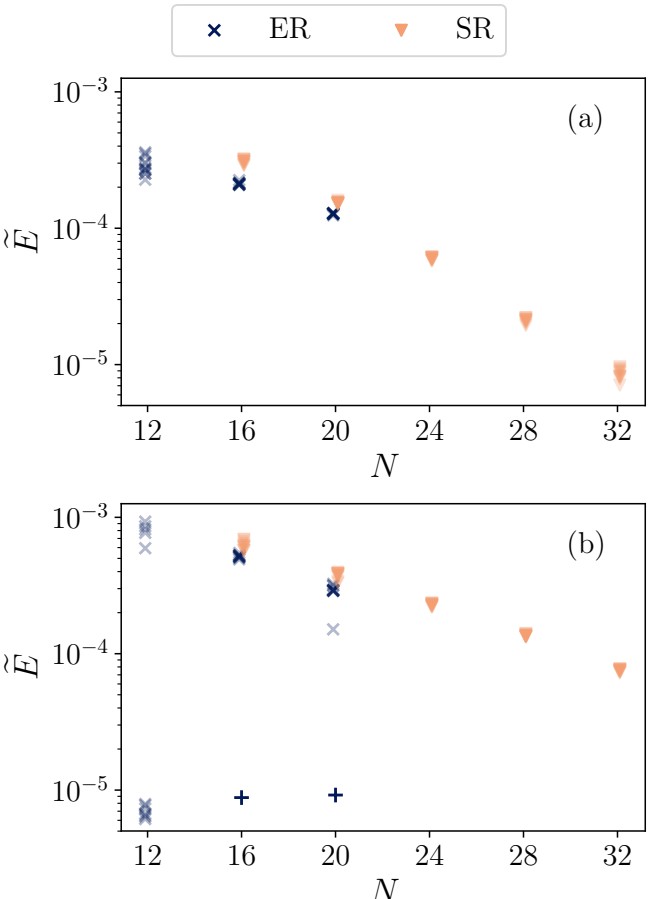

Figure 11:    Converged normalized energies for $H_{\text{tXYZ}}^{\diamond}$ with parameters (a) $(a, b) = (0.81, 1.31)$ and (b) $(0.89, 1.39)$ using the ER and SR. Plus markers $(+)$ in (b) indicate converged energies from the Hamiltonian annealing.

where we set $a = b = C$. Applying Pauli-$Z$ gates to all even sites (the sign rule) yields

$$H_{tXYZ}^{*} = C \sum_{i} (\sigma_i^x \sigma_{i+1}^x + \sigma_i^y \sigma_{i+1}^y - 1/C \sigma_i^z \sigma_{i+1}^z)$$
$$- (\sigma_i^x \sigma_{i+2}^x + \sigma_i^y \sigma_{i+2}^y + 1/C \sigma_i^z \sigma_{i+2}^z). \tag{19}$$

The Hamiltonian has $U(1)$ symmetry, so it preserves the total magnetization $J_z = \sum_i \sigma_i^z$. When $C < 1$, it is not difficult to see that the nearest-neighbor and the next-nearest-neighbor couplings both prefer the aligned states, which suggests that $|0\rangle^{\otimes N}$ and $|1\rangle^{\otimes N}$ are the degenerated ground sates. This implies that the $\mathbb{Z}_2$ symmetry along the $z$-axis is broken. In contrast, the Hamiltonian with $C > 1$ prefers $J_z = 0$ thus $\mathbb{Z}_2$ symmetry $\sigma_z \leftrightarrow -\sigma_z$ is restored.

Fig. 2 in the main text shows how the phases extend off $a = b$. Especially, there are two phase transitions along the line segment from $A = (0.25, 0.75)$ to $E = (1.25, 1.75)$ that we have simulated vQMC. To locate the second transition point, we calculate $d^2 E(\theta)/d\theta^2$ where $E(\theta)$ is the ground state energy of $H_{\text{tXYZ}}$ when parameters. We use paths $P_1$ and $P_2$ depicted in Fig. 6 of the main text. As path $P_1$ is a subset of $\overline{AE}$ we simulated the vQMC, we can locate the point $C_2$ in Fig. 2 using the result. Path $P_2$ is additionally considered to confirm that our phase diagram Fig. 2 indeed captures phase $\Lambda$ correctly.

To be precise, we set $(a,b) = (0.7, 1.2) + \theta(1,1)$ for $P_1$ and $(a,b) = (0.55, 1.45) + \theta(1,1)$ for $P_2$. The results from $P_1$ and $P_2$ are shown in Fig. 10(a) and (b), respectively. We see that there are two local maxima in the second derivatives along each path and the distance between two maxima increases as the path is moving away from the $a = b = 1$ point. This confirms the existence of an intermediate phase $\Lambda$ as depicted in Fig. 2.

## E   Scaling of converged energies in Phase II of the twisted XYZ model

In the main text, the converged energies along the path $\overline{C_2 D}$ with the ER and the SR have shown a different shape when $H_{\text{tXYZ}}^{\Diamond}$ is used. In this Appendix we show that this is a finite size effect.

To study this case more carefully, we simulate the ER for system sizes $N = [12, 16, 20]$ and the SR for system sizes $N = [16, 20, 24, 28, 32]$ using the two points indicated by arrows in Fig. 4 which are $(a,b) = (0.81, 1.31)$ and $(0.89, 1.39)$. The results in Fig. 11 show that converged energies from the SR decrease exponentially with $N$ which suggests that the optimizing problem vanishes in a large $N$ and we can solve the Hamiltonian using the SR.

One possible explanation for how such a good convergence is obtained for large $N$ is overparametrization. In classical ML, it is known that overparmterized networks optimize better [42]. Likewise, if the ground state is already sufficiently described by a small number of parameters, we expect that one can obtain a better convergence by increasing the network's parameter. As the number of parameters of our network increases quadratically as $\sim \alpha N^2$, this scenario is plausible if the number of parameters to describe the ground state scales slower than this. However, we leave a detailed investigation of this conjecture for future work.

## F   Sampling problems from non-stoquastic basis and from heavy tail distributions

In the main text, we have observed two different types of sampling problems. The first one appeared when we used a Hamiltonian in a non-stoquastic basis (e.g. the XXZ model in the antiferomagnetic phase without the sign rule). When this happened, converged energies of most of the SR instances are clustered far above the ground state energy ($\widetilde{E} > 10^{-2}$) and it persists regardless of the system size and the number of samples. We next observed a seemingly similar problem when solving the twisted XYZ model in Phase II using the stoquastic version of the Hamiltonian. However, the converged energies in this case were much closer to the ground state energy ($\widetilde{E} < 10^{-3}$ for all instances) and increasing the number of samples helped for $N \leq 28$. In this Appendix, we compare the learning curves in both models and show that the problems indeed have different profiles.

For comparison, we use the XXZ model with $\Delta = 1.5$ and the twisted XYZ model with $(a,b) = (1.23, 1.73)$. We plot 4 randomly chosen learning curves when $N = 28$ and $N = 32$ for both models in Fig. 12. One can easily distinguish the learning curves from the XXZ model without the sign rule Fig. 12(a), (b) and the twisted XYZ model with heavy tail distributions (c), (d). Specifically, Fig. 12(a) and (b) clearly show that the sampling problem enters in the middle of learning process and ruins the learning process when we use a non-stoquastic basis. In contrast, the learning curves shown in Fig. 12(c) and (d)

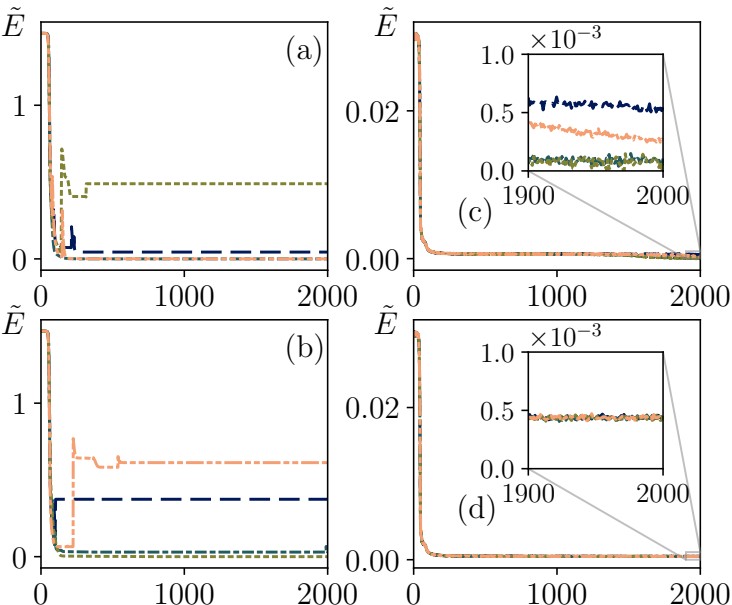

Figure 12: Four randomly chosen learning curves from (a, b) the XXZ model at $\Delta = 1.5$ without the sign rule and (c, d) the twisted XYZ model deep in phase II when $(a, b) = (1.23, 1.73)$. The system size $N = 28$ is used for (a) and (c), and $N = 32$ is used for (b) and (d).

first approach local minima that is near $\tilde{E} \approx 0.5 \times 10^{-3}$ and stay there for more than 1000 epochs. After that, some of the instances succeed in converging to better minima when $N = 28$. This behavior has also been seen in quantum chemical Hamiltonians [23]. These examples show that the origin of the two sampling problems are different.

## G Expressive power of the RBM for phase Λ of the twisted XYZ model

In Sec. 4.2.5, we have seen that errors from the ER increases as the parameter moves deeper in phase Λ of the twisted XYZ model. In this Appendix, we study a scaling behavior of errors when the Hamiltonian parameter is deep in phase Λ. Especially, we use the point $K$ in Fig. 6 which is $(a, b) = (0.4, 3.0)$ and the Hamiltonian $H_{\text{tXYZ}}^{\star}$ which reported the best converged energies from the ER.

In contrast to the deep non-stoquastic phase of the $J_1$-$J_2$ model that we studied in Appendix B, our SR results in Fig. 13(a) suffer from the *convergence* problem. Still, the normalized energies seem to converge to a positive value $> 10^{-4}$ with $N$ even when we take the best results for each $N$. On the other hand, Fig. 13(b) shows the converged energies from the ER for varying $\alpha = M/N$ when $N = 20$. As in the $J_1$-$J_2$ model case [Fig. 8(b)], the improvements are getting marginal as $\alpha$ increases. From these observations, we confirm that the RBM does not represent the ground states in phase Λ faithfully.

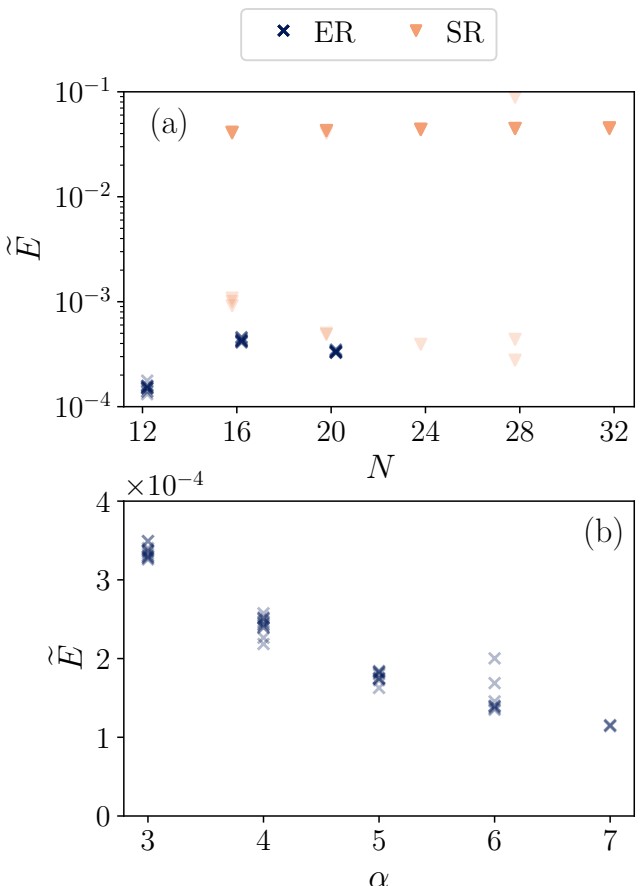

Figure 13: Converged normalized energies of the twisted XYZ model at point $(a, b) = (0.4, 3.0)$ which is deep in phase $\Lambda$. The transformed Hamiltonian $H_{\text{tXYZ}}^{\star}$ is used. We plot (a) the results from the ER and SR as a function of $N$ when $\alpha = 3$ and (b) the ER result as a function of $\alpha$ when $N = 20$.

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
