# Peer review of "Are neural quantum states good at solving non-stoquastic spin Hamiltonians?"

_SciPost Physics_

## Round 1 · Referee Report · Anonymous · 2021-3-8

Strengths

1-comprehensive review of the role of stoquasticity in RBM-based variational Monte Carlo
2-identification of three distinct mechanisms for learning failure
3-novel techniques to distinguish these mechanisms in numerical experiments

Weaknesses

1-narrow focus on RBM wave functions
2-overly general claims for the effect of the "sign problem" on other neural network ansätze

Report

In this paper, the authors study the ability of variational Monte Carlo (VMC) using restricted Boltzmann machine (RBM) wave functions to find the ground states of spin chains with sign-problem-free (stoquastic) and non-stoquastic Hamiltonians and discover "deep non-stoquastic phases" in which RBMs tend to fail to find an accurate ground state.

The authors identify three mechanisms by which such learning can fail: inability of the ansatz to represent the ground state; failure to converge to an otherwise representable ground state; failure of sampling to reconstruct the expectation values used in the learning protocol. While each of these failure mechanisms had already appeared in the literature, they introduce ingenious techniques to separate them from one another. Using careful and systematic numerical experiments, they demonstrate that RBMs fall short on each count when the Hamiltonian is not phase-connected to a stoquastic point, which they call a "deep non-stoquastic phase;" on the other hand, VMC on non-stoquastic Hamiltonians that share a phase with stoquastic ones succeeds despite their sign problem. Together with the results of https://www.nature.com/articles/s41467-017-00705-2 and https://arxiv.org/abs/2011.08572, these results delineate the range of usefulness of RBM wave functions, which is crucial for reliable physical applications.

In the body of their work, the authors maintain a narrow focus on RBM wave functions. While this is a limitation considering the more elaborate neural quantum states (NQS) that have appeared in the literature (e.g., Refs. 12-15), the results are still significant on account of the wide usage of RBM ansätze, motivated by little more than precedent.
However, the introductory and concluding sections of the work attempt to generalise their findings to other NQS states, even where this does not seem justified given the existing literature. Specific examples:
* In the abstract: "Based on anecdotal evidence, it has been claimed [NQS] are insensitive to the sign problem." In fact, all pieces of literature that set out to study non-stoquastic problems with purely NQS methods (i.e., without relying on Gutzwiller-projected ansätze; see Refs. 12,13,15,17,22,38,41, and a host of other works) have seen difficulties that arose from the sign problem.
* §5 discusses Ref. 17 suggesting that it failed to account for the insufficient expressivity of the network. In fact, it explicitly shows that the network used there is able to learn the sign structure accurately if the entire Hilbert space is used in the training, analogous to the ER protocol of this work. This is not surprising as their networks are more flexible and hence expressive than RBMs, and rather underlines that for a generic deep network, expressivity is unlikely to be an issue. The same applies to the subsequent note on Ref. 38.

A very interesting point raised by the authors is a sampling problem observed in the stoquastic Phase II of the XYZ chain, which is not due to the sign problem, but rather a heavy-tailed distribution of quantum probabilities. It is unfortunate this point is not studied further, as it is a property of the ground state itself, so it has implications for VMC in general. I hope they will follow it up either in a revision of this paper or in future work.

In summary, this work adds significantly to our understanding of the abilities and limitations of RBM wave functions and introduces useful techniques for systematically studying the performance of other VMC ansätze, which merits publication in SciPost Physics. However, I ask the authors to revise the introductory and concluding parts of the manuscripts to bring their claims in line with what is supported by their actual results. I list specific suggestions and some more minor comments and questions in the "Requested changes" section.

Requested changes

The requests are broken into three blocks: essential revision; improvements to content; improvements to form

1-Consider changing the title to a statement that is explicitly about RBMs
2-Change the discussion of Refs. 17 and 38 in §5 observing that failures in expressivity are less likely to play a role for deep networks, as indeed found by Ref. 17
3-Revise the sentence in the abstract that I highlighted in the report

4-Consider turning §5 into a conclusion section, explicitly collecting the key observations from the body of the paper (or perhaps moving them from §1)
5-Clarify the description of methods towards the end of §2. In particular
5a-The sentence "In the case of Hamiltonians adiabatically connected to a stoquastic phase, we then change the basis to see whether the sampling problem persists" is confusing - if it is already in a stoquastic phase, why change the basis?
5b-Explain what you mean by the annealing procedure where it is first mentioned
5c-Give more technical details (libraries used, initialisation, epoch numbers, step sizes, etc.) in an appendix to aid reproducibility.
6-Describe the phase diagram of the XXZ chain in §3.1 to a similar level of detail as that of the J1-J2 Heisenberg chain.
7-At the bottom of page 5, what do you mean by "different temperatures" for a ground-state simulation?
8-Near the Majumdar-Ghosh point, the uncured Hamiltonian seems to outperform the cured one in SR, even though the exact ground state is known to obey the sign rule. Have you studied why this is?
9-In §4.1 and the caption of Figure 2, clarify the role of the transformed Hamiltonians. It would be instructive to mention where the untransformed Hamiltonian is stoquastic; in the caption, it currently reads like that is the case for the whole first and third quadrants, which is very confusing
10-Fig. 3 shows that Phase II breaks the $Z_2$ symmetry in both $\sigma^x$ and $\sigma^y$, but this is not mentioned either in the caption or the main text. Why? What is known about the ground state degeneracy in this phase? Could this have some bearing on the "heavy-tail" sampling problem?
11-§4.2.2 says that both transformed Hamiltonians working better than the untransformed one reinforces Observation 1. This only makes sense for $H^\star$, but not for $H^\diamond$, which is not stoquastic in this regime, either. Could you clarify this?
12-In Appendix B, the error of ER/SR is contrasted with the energy of the lowest excited state. (a) The latter is said to scale as 1/N - how does this square with the state being gapped? (b) Consider adding the energy of the lowest excited state to Fig. 8 to help assess how well the VMC energies approach the ground state.

13-Figures 1 and 4 are hard to read.
13a-Add tics on the y-axis every decade (maybe smaller than the ones you already have) on both sides to aid comparing ER and SR energies.
13b-In Fig. 4, the purple colour can be confused with both the blue and the orange, especially at low opacity. Consider using another colour (green?)
13c-The dotted purple line is very easy to confuse with data points. Consider using a solid line
14-Remove the single line of main text from Page 9
15-Proofread the text thoroughly for typos, grammar, and unclear explanations (I tried to flag up most of the latter above)

---

## Round 1 · Referee Report · Anonymous · 2021-3-21

Strengths

1 - Clearly written;
2 - Introduces a systematic approach to an open and interesting problem;
3 - Presents a systematic analysis of relevant models.

Weaknesses

1 - It does not meet any of the required expectations;
2 - No definitive clarification of the open problem.

Report

The manuscript presents a solid analysis of a timing and important problem, namely, it analyses the performances of the neural network approach to the study of many-body quantum systems. Recently, following a seminal paper of Carleo and Troyer, it has been argued that neural networks are strong candidates to be used as ansatzes to describe ground states of many-body quantum Hamiltonians.

This paper presents an unbiased analysis of such an approach, concentrating on Restricted Boltzmann Machines and some particular challenging Hamiltonians. Throughout the manuscript, the authors clearly individuate and describe three possible failures or challenges of the method (sampling, convergence and expressitivity of the ansatz) and explore their relations and relate them with the stoquasticity of the Hamiltonian and of their different phases. In particular, the authors present their analyses on the Heisenberg model, the J1-J2 model, and the twisted XYZ model.

The result of the work presented here are three observations that set the stage for further analysis and possibly the development of a full theory and of our complete understanding of such an approach. I liked the manuscript that, despite some minor issues described below, is clearly written and presents a highly technical issue in a clear and fair way. I highly appreciate also the honesty of the authors, for example, when they explicitly write "we do not have a good explanation for this behavior". I also think that this manuscript will serve as a useful starting point for further discussion and as a reference point for other in-depth analyses.

However, the presented results do not match any of the expectations enumerated by the editorial board: no groundbreaking discovery nor breakthrough is presented, or new pathways are introduced. The work is also not connecting different areas, actually, this is a very interesting solid technical work that most probably will interest only experts in the (sub)field working on the topic. For this reason, I recommend publication in Sci Physcis Core once the authors have considered the following comments.

Requested changes

1 - To improve readability to a wider audience, I suggest the authors to briefly recap (maybe in section 2) the problem that non-stoquasticity introduces in vQMC approaches.

2- Similarly, a few sentences on MCMC and swap update rule could improve readability.

3 - Figures are in general very dense: in Fig. 2 shading with different colors the regions with different behavior might help the reader.

4 - Fig 2: I do not understand why the authors change both the model AND the size N from panels a- to c-d.
Changing two properties might suggest that something else is happening, and I do not understand why they cannot keep N constant.

5 - Fig.3: The authors claim to see a logarithmic scaling,
by there is no figure to support this statement. Moreover, nowhere it is described which entanglement entropy they are studying: I suppose they are cutting the chain in the middle, but they should define explicitly what they do.

6 - The authors keep referring to values of the order of 10^-5 as converged, while they are not satisfied with values of the order of 10^-3. While I understand and probably share their statements, these two values are completely arbitrary. I think that this choice deserves some comments or a discussion (e.g., are the results stable changing these thresholds?).

---

## Round 1 · Referee Report · Anonymous · 2021-3-22

Report

The paper presents a discussion of some simulations realized on one-dimensional spin models, using Restricted Boltzmann Machines (RBM) with complex parameters as a variational ansatz.
The authors use these results to draw some general conclusions on what they identify to be three failure mechanisms of RBMs in these cases.

While it is certainly very important to provide some detailed examples and give an idea of what are the failure modes of a given method, it seems to me that the present manuscript is of insufficient quality in many respects (see my detailed comments below) and should not be granted publication, unless substantially and thoroughly revised.

The conclusions that are drawn here are not strongly substantiated by numerical results and go against a series of well-reproducible results from the past years obtained by several independent groups worldwide.

The paper for example falls short in addressing the fact that there exist results (even with simple RBMs) that are state of the art in the simulation of challenging frustrated models. See for example [https://arxiv.org/abs/2009.14777] for one of the latest.
In this sense, it is unclear what "anecdotal evidence" in the abstract means and what is implied in terms of the significance of a large body of independent works.

The topic studied in this work also goes along similar lines of several other
works who have investigate the role of non-stoquasticity and frustration.
For example, ref [13,17,38] to name a few.
Ref [38] is certainly the closest to this study, and it is remarkable that it draws significantly different conclusions than the present study. This fact alone should be considered as a red flag for the generality of the conclusions the authors draw here. What is identified in this work reflects some very well known limitation of Markov Chain Monte Carlo (the so-called sampling problem, in this work) that exist also for classical systems, as well as some limitations of high-dimensional optimization, that exist in a huge variety of machine learning applications. These issues are not specifically restricted to applications of neural quantum states, but are much broader in computational science. Most importantly, there are known and effective ways to circumvent them, as demonstrated by the succesfull applications of these methods to frustrated models, by other groups.

In this respect, reference [38] correctly identifies the rugged nature of the optimization landscape as **the** main issue encountered when optimizing neural quantum states. In this work there is no sufficient evidence to refute the conclusion of [38], and also
of many other existing works who have demonstrated the high representability power of neural network states, see [17] for example.

More specifically, the 3 main failure modes are addressed below are either
already known/ have possible solutions, or are more general features
appearing in many other related computational techniques.

1. "The sampling problem" is a very specific issue the authors identify here, with apparently surprising results. It is found that there exist regimes where the probability density associated with the square of the wave function is such that it becomes hard to sample using the simple Monte Carlo strategies adopted. It can be argued there this has nothing to do with a) the gap of the quantum hamiltonian b) the sign/non-stoquastic problem.

For example, it is very well known that MCMC can fail even in very simple models such as the classical Ising model. Most importantly, the sampling problem is fully and effectively solved for neural quantum states by autoregressive or similar networks [14,15] or by using more advanced sampling techniques than the basic ones adopted in the current work.

2. The "convergence" problem is in my opinion the only non-trivial issue raised here, and which is discussed carefully and in depth in Ref. 38.
However, any non-trivial high-dimensional optimization scheme suffers from this issue, and failure modes of all optimization schemes can be very easily found. The whole body of modern machine learning is concerned with solving this very optimization problem, and using more expressive, over-parameterized networks (so, well beyond the tiny RBM considered
here) is a solution adopted successfully by several works.
In the context of neural quantum states, the optimization problem is often alleviated by changing parameterizations and using a training of the sign as done in Ref. [13]. Have the authors actually tried that optimization scheme or alternative parameterizations of the complex amplitudes?

3. The "expressivity issue" is something that has already been well studied in literature, see [14,15,17,38] for example. These studies show that deep neural quantum states have superior performance than shallow RBMs [14,15]. Also, it is found that expressivity is not an issue for deep networks,
also in strongly frustrated phases [17,38].
Seen from a pure machine learning perspective it is also highly expected that a shallow neural network performs worse than a deep network.
Using experiments on shallow neural networks only, as done in this work, to draw general conclusions on the expressive power of neural quantum states is misleading and contradicts mounting evidence from several other studies who have taken the time and the effort to implement succesfully deep neural quantum states.

Furthermore, there are also two other main issues :

4.The results are not reproducible, since there is no code or reference implementation used.

5. The Exact Reconfiguraiton approach does not have noise in the gradients, thus it is intrinsically likely to get stuck in local minima.
For this reason, all results based on Exact Reconfiguration are not particularly telling of the actual optimization issues involved, who instead use stochastic gradients and are more resilient to local minima. As explained in Ref [38] there is a subtle interplay between parameterization, complex-valued parameters, activation functions and several other things that need to be taken carefully into account.

In conclusion, I believe that this work should be significantly rewritten and make statements that are more in line with what actually actually achieved in the numerical simulations at hand. Also, it is a bit disconcerting that a great deal of conclusions of this work are based on the fact that some of the optimization instances fail to get accuracies higher than 10^-3, which is by many metrics a very high accuracy for a simple, shallow RBM ansatz.

Requested changes

1. The sampling problem is most likely due to an ergodicity issue in the monte carlo sampling.
There is no effort done here to characterize the sampling problem and alleviate it using different sampling schemes. I would suggest to use other sampling approaches or, at the very least, rely on an exact sampler (for the small instances this is always possible, if all the states in the Hilbert space are enumerated).

2. In the "deep non-stoquastic phase", and for the cases where the RBM performs the worst what is the initial state orthogonal or almost orthogonal to the exact ground state? If so, a better initialization strategy for the variational parameters has to be adopted.

3. The title should really reflect the fact that the authors are addressing only shallow RBM neural quantum states and that their conclusions are strongly limited in scope as such.

4. "Based on anecdotal evidence, it has been claimed repeatedly in the
literature that neural network quantum state simulations are insensitive to the sign problem". This kind of statements that are factually contradicted by many other works who have studied the same problem should be removed, also because they are not very helpful for an healthy scientific debate.

4."The relevance of stoquasticity for the vQMC is far less explored, despite the fact that this method and its variants are often advertized as solving the sign problem [11]" . From our understanding, that paper does not mention/advertise that they solve the sign problem.

5. The note 1 "Still, one may see a sampling error if a target probability distribution cannot be well approximated with a finite number of samples." is incorrect. It is known that if one can sample from the exact
distribution can obtain any quantum expectation value of a local observable to arbitrary precision, see [https://arxiv.org/abs/0911.1624]

6. "Observation 3. Sampling is stable along the learning path when the Hamiltonian is stoquastic or phase connected to a stoquastic Hamiltonian." Again, sampling issues appear also in the classical domain and are strictly unrelated to stoquasticity.

7. "we explicitly show that one may encounter such a problem even in solving a one dimensional stoquastic system which one would expect..."
It is not expected that all classical models are efficiently samplable (again, no sign problem involved, just the fact that if you have a glassy landscape it is hard to find/sample from the ground state for any known computational method to the humankind).

8. "This is quite surprising, since the MCMC simply uses the ratio between two probability densities ..., which is sign invariant. " I would certainly not emphasize what is "surprising" or not, since this is again highly dependent on the taste of the reader. On the contrary, I would
argue that any reader with a reasonable experience in classical Monte Carlo would not find surprising at all that, say, MCMC fails on the classical Ising model at low temperatures, despite "the absence of sign problem" there. In that context, would the authors draw the conclusion that "it is surprising that convergence is not observed, despite the fact
that the Boltzmann weight has no sign problem" ? if the answer to this question is no, then I would strongly suggest to entirely rethink the nature of their conclusions on this point.

For example, it is highly likely that the sampling problem encountered in the non-sign rotated XXZ model is due to the fact that local updates in MCMC are not enough to generate ergodic samples.
It would be instructive if the authors tried sampling from the exact distribution (using direct sampling on small systems, enumerating the full state space) and compare to what they get in the other cases.

9. In Fig 1 c it is argued that there is an expressive power issues because "some instances" have errors above 10^-4.
This seems to me by far the weakest of the conclusions drawn by the authors, since there is a typical error of 10^-3 even in Fig. 1 (d), the sampled case.

10. "..of the network is insufficient for describing the ground state
even though the system is gapped" why is it expected that an RBM can more easily describe a gapped state?

11. "using the MCMC with the swap update rule." What is the "swap update rule"? is this parallel tempering?

---

## Editorial Decision

editor-in-charge_assigned